# Changes in endolysosomal organization define a pre-degenerative state in the *crumbs* mutant *Drosophila* retina

**Rachel S. Kraut** ![ORCID] *, **Elisabeth Knust**

Max Planck Institute of Molecular Cell Biology and Genetics, Pfotenhauerstrasse, Dresden, Germany

* rkraut1000@gmail.com

**Data Availability Statement:** All relevant data are within the manuscript and its Supporting Information files.

**Funding:** This work was supported by the Max-Planck Society (https://www.mpg.de/en). The

## Abstract

Mutations in the epithelial polarity gene *crumbs* (*crb*) lead to retinal degeneration in *Drosophila* and in humans. The overall morphology of the retina and its deterioration in *Drosophila crb* mutants has been well-characterized, but the cell biological origin of the degeneration is not well understood. Degenerative conditions in the retina and elsewhere in the nervous system often involve defects in degradative intracellular trafficking pathways. So far, however, effects of *crb* on the endolysosomal system, or on the spatial organization of these compartments in photoreceptor cells have not been described. We therefore asked whether photoreceptors in *crb* mutants exhibit alterations in endolysosomal compartments under pre-degenerative conditions, where the retina is still morphologically intact. Data presented here show that, already well before the onset of degeneration, Arl8, Rab7, and Atg8-carrying endolysosomal and autophagosomal compartments undergo changes in morphology and positioning with respect to each other in *crb* mutant retinas. We propose that these changes may be early signs of the degeneration-prone condition in *crb* retinas.

## Introduction

Intracellular protein trafficking is essential for the maintenance of cell and tissue homeostasis. A multitude of functions are dependent on intracellular transport, including signal transduction, secretion or membrane remodelling, to mention just a few. Therefore, it is not surprising that impaired trafficking, induced by aging, environmental factors or mutations in genes encoding important components involved in trafficking, is associated with numerous detrimental human diseases. These include neurodegenerative diseases in particular, characterized by the progressive loss of neuronal function, such as Alzheimer's and Parkinson's disease, or retinal degeneration, leading to blindness [1, 2]. Important insight into the regulation of intracellular trafficking has been obtained by studies in model organisms, notably the fruit fly *Drosophila melanogaster*. The enormous number of mutants collected over the years, the sophisticated genetic and molecular toolkit as well as the ease of performing unbiased genetic screens [3, 4] have uncovered novel molecular pathways and identified a plethora of genes safeguarding neuronal homeostasis, including those regulating intracellular trafficking [5, 6].

funders had no role in study design, data collection and analysis, decision to publish, or preparation of the manuscript.

**Competing interests:** The authors have declared that no competing interests exist.

Importantly, roughly 75% of genes associated with human diseases have functional orthologues in the fly genome, many of which are associated with neurodegeneration [7, 8]. The fly eye is an ideal organ to study the basis of neurodegeneration and to identify the players involved: it is a non-essential organ, in which genetic mosaics can be induced, and is easily accessible to cell biological and electrophysiological analyses [3, 4, 9]. In addition, retinal photoreceptor cells are heavily reliant on the membrane trafficking system due to the need for constant renewal and repair of light-induced damage to membranes, and hence are very sensitive to any impairment in trafficking.

Phototransduction in the *Drosophila* retina takes place in a specialized organelle called the rhabdomere. It is formed at the apical surface of each photoreceptor by ~ 30,000 densely packed microvilli [10, 11]. Maintenance of this organelle depends on extensive membrane turnover to constantly replenish molecules of the microvillar membrane, including the light sensitive molecule rhodopsin. This process is particularly challenged during light exposure due to continual endocytosis of light-activated rhodopsin, followed by recycling to the rhabdomere or by degradation via the endolysosomal trafficking pathway [12, 13]. These observations underscore the importance of all trafficking compartments, including those involved in autophagy, degradation and recycling. Failure in any step of these pathways often results in aberrant rhodopsin accumulation in a late endosomal, Rab7-positive compartment, and represents one cause of retinal degeneration [14–16].

One of the central players in the breakdown of biomolecules is the lysosome. Lysosomes are dynamic, membrane-bound organelles, which were initially identified as sites for degradation of intracellular components. There is increasing evidence that lysosomes are similarly important for a plethora of other functions, including metabolic signalling and plasma membrane repair [17]. Lysosomal functions are closely associated with autophagy, a conserved cellular process required for degradation and recycling of nutrients and aged or damaged organelles, whereby cellular components are enclosed in double-membrane vesicles, the autophagosome [18]. Fusion of the autophagosome with the lysosome results in formation of the autophagolysosome, in which biomolecules are degraded and/or recycled to support energy production [19].

Lysosomes are not static entities, but highly dynamic structures. They can change their intracellular position by moving bidirectionally along microtubules, and their position within a cell can have a major effect on their function [20–22]. Depending on the effector bound, late endolysosomes associated with the small GTPase Rab7 can be transported to either the microtubule minus or plus end by interactions with dynein or kinesin, respectively. Plus-end directed transport of endolysosomes is facilitated by Arl8, a small GTPase of the Arf (ADP ribosylation factor) family, which seems to engage in a "tug of war" with Rab7, and together with Rab7 can lead to tubulation of lysosomes [23–27]. In addition to Arl8 and Rab7, the retromer and its interactions with the lysosome play important roles in the degradative process important in maintaining photoreceptor homeostasis and integrity [16, 28].

Besides genes encoding components of the trafficking machinery such as the retromer, many other fly genes are required to ensure photoreceptor survival [29]. One of these is *crumbs* (*crb*), mutations in which result in light-dependent retinal degeneration [30–32]. Similarly, mutations in one of the three human orthologues, *CRB1*, result in Retinitis pigmentosa (RP12) and Leber congenital amaurosis, two of the most severe retinal dystrophies associated with blindness [33–35]. Crb is a transmembrane protein expressed on apical membranes of epithelial cells and in photoreceptor membranes in the fly [31, 36]. It possesses a large extracellular domain with EGF and laminin-like repeats, and a short intracellular domain that organizes the so-called Crb-complex of interacting proteins [37]. Retinal degeneration in *crb* mutant photoreceptors is strongly attenuated when flies are kept in the dark, or when raised and kept on

food lacking vitamin A, a cofactor necessary for functional rhodopsin [31]. In addition, lack of *crb* impairs MyosinV-dependent rhodopsin trafficking to the rhabdomere [36]. We recently showed that the retina of a hypomorphic *crb* allele, *crb*[P13A9], degenerates upon constant exposure to light [32]. The retina of this allele lacks the morphological defects in cell shape observed in other *crb* alleles [31, 37, 38], suggesting that retinal degeneration associated with lack of *crb* is independent of its developmental phenotype. These observations, together with the fact that *crb*[P13A9] mutant flies are homozygous viable, makes this allele ideally suited to address the question to what extent a reduction of *crb* function affects the endolysosomal system and thereby could make photoreceptors more prone to damage by light.

Here, we show that photoreceptors in hypomorphic *crb*[P13A9] mutant flies exhibit alterations in endolysosomal compartments under normal, non-degenerative light conditions, in which the overall morphology of the retina is still intact. Moreover, the spatial relationships between different components associated with endolysosomal and autophagosomal compartments is altered in *crb* mutant photoreceptors. From these results we hypothesize that spatial changes in endolysosomal organization could define a "pre-degenerative condition", which makes mutant photoreceptors prone to degeneration upon light stress.

## Materials and methods

### Fly stocks and culture

Fly stocks used in this study were: *w** [32] (used as control), *w; crb*[P13A9] [32], *w; Rh1*-Gal4 [39], *w;* UAS-LAMP-GFP (a gift of H. Krämer;[40]), GeneTrap CG6707 [41], *w**; UAS-Rab7-GFP (a gift of M. Gonzalez-Gaitan;[42]), *yw;otd*[1.6]-Gal4 pWIZ8 (UAS-*w*[RNAi]) (a gift of T. Cook; [43], *yw;* UAS-*spinster*-RFP (a gift of S. Sweeney; [44]), UAS-*Atg8a*-mCherry (a gift of T. Neufeld; [45]). For expression of transgenes encoding fluorescent protein (FP) in the *crb* background, the stock *w; Rh1*-Gal4; *crb*[P13A9] was crossed to UAS-FP; *crb*[P13A9], and *Rh1*-Gal4/ UAS-FP; *crb*[P13A9] adults were selected. Unless otherwise stated, flies were raised on standard cornmeal medium, and kept at 25˚C either on a 12h light (2100–2500 lux), 12h dark (0 lux) cycle, or in constant light at ~2100–2500 lux, or in complete darkness during pupariation and molting, until the time of dissection. For experiments with flies raised on vitamin A-depleted food, larvae were raised and flies were maintained on carotenoid-depleted food [31, 46].

### Dissection and preparation of retinas

Adult females were decapitated under $CO_2$ and heads were bisected between the eyes with a razor blade before fixing them in cold, freshly prepared 4% formaldehyde solution for 2h on ice. Eyes of dark-reared flies were prepared in dim room-light without microscope illumination, and fixation was carried out in the dark. Preparation and fixation of retinas from flies raised in constant light was done under bright light. For 12 h light/12 h dark experiments, flies were dissected at roughly the same time of day, i.e. after ~6h of light exposure. Fixed eyes were washed in PBS/0.3% TritonX-100 (PBTx) and dissected in phosphate-buffered saline (PBS), using an insect pin with the tip bent, so that intact retinas could be scooped out of the lens. 8–10 retinas per condition were transferred individually using a P20 tip into glass wells with PBTx and blocked in 5% normal goat serum 30 min at room temperature. For VitaminA-depletion experiments, flies were hatched and reared completely on carotenoid-free medium [46].

### Immunohistochemistry

Retinas were incubated for at least 15h at 4˚C in primary antibodies diluted in PBTx with 1% normal goat serum. Primary antibodies used in this study were: rat anti-Crb (batch 2.8; [47],

1:1000); rabbit polyclonal anti-Arl8 (1:200)[48]; Developmental Studies Hybridoma Bank (DSHB), Univ. Iowa); mouse monoclonal (MAb) anti-alpha Sodium-potassium ATPase (1:200)(AbCam #7671); mouse MAb anti-Rab7 (1:5)([48]; DSHB); rabbit anti-GFP (1:1000, Invitrogen/ThermoFisher, Cat. no. A11122); mouse MAb anti-mCherry (1:1000, MPI mono-clonal antibody facility, Dresden); mouse anti-Rhodopsin1 (1:200, Millipore, Cat. no. MAB5356). Alexa 488-, 568-, and 647-coupled secondary antibodies (ThermoFisher) were used at 1:1000 dilution, and DyLight 405 secondary antibody (Jackson ImmunoResearch) at 1:500. Alexa680-coupled Phalloidin (ThermoFisher, Cat. no. A22286) was used at 1:200 to detect F-actin, together with secondary antibodies.

### Microscopy and image analysis

Retinas were mounted in Vectashield (VectaLabs), under sealed coverslips supported with two streaks of nail polish to avoid flattening of the tissue. High-resolution imaging was done on a Zeiss LSM 880 Airy microscope with a 63x/1.4 NA lens. Imaging for quantification of spot sizes and intensities and for co-localization of Arl8-Rab7 was done on a Yokogawa Spinning Disk confocal microscope with a 100x/1.4 NA Silicon immersion lens, at 117 nm pixel dimensions. Laser and exposure settings were kept constant and below-saturation for all experiments.

The procedure for quantification of size and intensity of Arl8 regions was as follows: for each retina or retina fragment, between 1–6 confocal image stacks of >200 images each, spanning the depth of the retina, were acquired with an Andor iXon Ultra 888 cooled EMCCD camera at 230 nm z-distance, and processed in Fiji before deconvolution with Huygens software. For each image slice at a minimum of 1 μm z-spacing, where longitudinally oriented ommatidia were present, judged by anti-Crb staining, Arl8-positive regions within the Region Of Interest (ROI) to be analyzed were determined by automatic thresholding to detect signal above background, yielding "spots". These spots, or Arl8 regions (outlined in beige), were defined as having an intensity over background (Signal–Noise, S-N) at least 5 standard deviations from the average intensity over the whole ROI. Average sizes and intensities of individual Arl8 spots (or other antigen of interest) within the region were recorded using a script based on the Find Maxima plug-in in Fiji (see S1 Script). The average for all spots in an entire stack was taken as one data point, and all data points for one experiment were normalized to the $w^*$ control for each experiment, yielding values for $w^*$ close to 1. Normalized data from up to three experiments were pooled for each graph. Control and $crb^{P13A9}$ samples were always processed in parallel and data were plotted in RStudio.

Pearson's correlation coefficient (R coloc) was calculated for Arl8/Rab7 from deconvolved spinning disk image pairs using the Coloc2 plugin in Fiji, with thresholding, on ROIs drawn around longitudinal regions of ommatidial organization, judging by anti-Crb staining. Optical image slices at ~1 μm intervals were analyzed in order to avoid redundant sampling. Atg8-m-Cherry co-localization with Arl8 was calculated from Airy processed confocal images, using the thresholded Manders' coefficient (tM2) for the Atg8-mCherry channel alone, based on the rationale that Arl8 would be found in all cells, whereas Atg8-mCherry was only driven in pho-toreceptors. Isolated Atg8-mCherry compartments were chosen as ROIs in the mCherry chan-nel, and these were then evaluated for tM2.

## Results

### Positioning of the Arl8-positive compartment in adult photoreceptor cells depends on light

We hypothesized that reduction of *Drosophila crb* in photoreceptor cells could render the cells susceptible to increased light stress, which ultimately results in retinal degeneration. Since

impaired trafficking in photoreceptors can result in retinal degeneration [16], which is often dependent on increased light intensity, we first aimed to determine the role of light on the organization/positioning of the endolysosomal system in wild-type photoreceptors. The Arf-like small GTPase Arl8, defined as a bona-fide lysosomal marker [24, 48], has been linked to various aspects of the endolysosomal system, such as intracellular transport and fusion with the autophagosome [23, 26]. We noted that the intracellular positioning of Arl8 within the retina was altered upon changes in light conditions. When flies were exposed to a 12 hour light/ 12 hour dark cycle ("light" = 2100–2500 lux, "dark" = 0 lux, defined as "normal light conditions" throughout the text), a proximo-distal alignment of the Arl8-positive compartments was evident in photoreceptors of eyes from the control strain (*w**) (Fig 1A and 1A'). Optical cross-sections (Fig 1B and 1B') revealed frequent positioning of a large, Arl8-positive compartment (red) nearer to the base of the rhabdomere, the actin-rich, light sensitive organelle of the photoreceptor (white), than to the basolateral surface of the photoreceptors (green; Fig 1B and 1B'). We used somewhat brighter light conditions than those described in previous publications [31], since alignment of Arl8 compartments in photoreceptors was only reliably observed under normal light conditions with higher intensity during light periods. This alignment was less pronounced in photoreceptors of flies that had been kept in constant intense light for 3 days (Fig 1C and 1C'), or kept in complete darkness during pupation and until 10 days after hatching (not shown). In addition, 3 days of constant light induced the formation of large Arl8-positive clusters, which were less frequently located close to the base of the rhabdomere, compared to normal light conditions (Fig 1C and 1D'). These results show that light conditions influence the spatial organization and positioning of the Arl8-positive lysosomal compartment in *w** control retinas.

To determine the identity of the Arl8-positive compartment in the retina, we co-labelled retinas with anti-Arl8 antibodies together with other markers of the endolysosomal trafficking pathway. The markers used included antibodies against endogenous Rab7 to label late endosomes [16, 42], and antibodies to detect overexpressed GFP-Rab7, mCherry-tagged Atg8 to label the autophago-lysosome [45], RFP-tagged Spinster (a gift of Sean Sweeney;[44]), a presumptive transporter involved in autophagosomal-lysosomal regeneration, and GFP-tagged Lysosome Associated Membrane Protein (LAMP), which is targeted to the lysosome [40] (S1A–S1E Fig). Retinal overexpression of fusion proteins was achieved using either *Rh1*-Gal4 or *Otd*-Gal4. We also used a protein trap line, carrying an insertion in CG6707, which encodes a lysosomal phosphatidyl-inositol bisphosphatase (PIP2-ase) (http://flybase.org/reports/ FBal0185170.html) and labels the lysosomes [41] (S1F Fig). The endogenous late endosomal protein Rab7, as well as Rab7-GFP expressed via Rh1-Gal4 in photoreceptors only (S1A and S1B Fig), are often found in Arl8-positive compartments, as is LysoPIP2-ase (S1F Fig). This contrasts with the autophagolysosomal markers LAMP-GFP, Atg8-mCherry, and Spinster-RFP, which occupy only partially overlapping or adjacent, but not identical compartments (S1C–S1E Fig). From these results we conclude that under normal light conditions, Arl8 is most closely associated with a late endosomal or endolysosomal compartment that also carries Rab7 and/or the lysosomal PIP2-ase CG6707.

## Arl8-positive compartments exhibit aberrant clustering in *crb* mutant retina under normal light conditions

The strong loss of function allele *crb^11A22* shows severe photoreceptor degeneration in response to light stress [30, 31]. However, the retina of these flies shows additional developmental defects due to impaired elongation of the rhabdomere during pupal stages [31, 38]. In contrast, flies homozygous mutant for the hypomorphic *crb^P13A9* allele are viable and fertile,

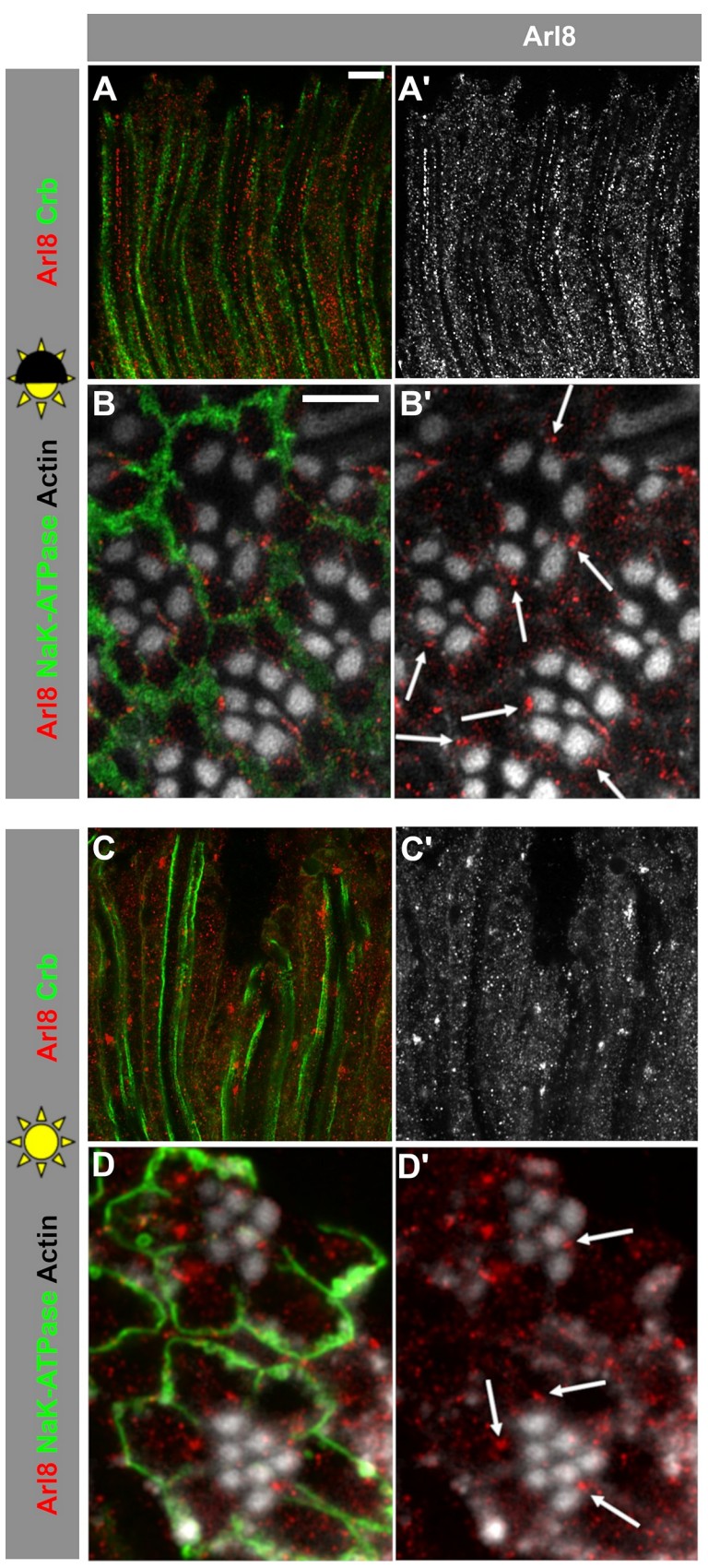

**Fig 1. The distribution of Arl8-positive lysosomal compartments depends on light conditions in *w*\* control photoreceptor cells.** (A-B') Lysosomal compartment marked with Arl8 in longitudinal (**A, A'**) and cross-sectional (**B, B'**) optical sections of *w*\* retinas after 5 days exposure to 12h light/ 12h dark (half sun).(**C-D'**) Lysosomal compartment marked with Arl8 in longitudinal (**C, C'**) and cross-sectional (**D, D'**) optical sections of *w*\* retinas after 3 days of constant bright light (full sun). Stalk membranes of photoreceptor cells in longitudinal sections are marked with anti-Crb (green; **A, C**), basolateral domains are labelled with anti-Na$^+$-K$^+$-ATPase in cross-sections (green; **B, D**), and actin-rich rhabdomeres are labelled with phalloidin (white; **B, B', D, D'**). Note that the alignment of the Arl8-positive vesicles along the base of the rhabdomeres under dark/light conditions is abolished under constant light. Scale bars: 5 μm.

and their retina does not exhibit any of the developmental or polarity defects caused by loss-of-function *crb* alleles. *crb*$^{P13A9}$ is a point mutation resulting in truncation of an eye-specific isoform of the Crb protein. *crb*$^{P13A9}$ only affects the isoform Crb_C, which contains an additional EGF repeat predicted to be heavily glycosylated (described in detail in [32]). Other Crb isoforms are normally expressed and localized in the retina. Yet, the retina of homozygous mutant *crb*$^{P13A9}$ flies undergoes degeneration under prolonged intense light conditions in a white-eyed (*white*, *w*\*) background [32]. Therefore, this allele is ideally suited to further study the effects linked to degeneration, but independent from any developmental effect.

When kept under normal light conditions (12 h light/12 h dark cycle), *crb*$^{P13A9}$ mutant retinas reveal a striking difference in the distribution of Arl8-positive compartments compared to that of control retinas kept under the same conditions. Arl8 compartments were no longer regularly aligned along the proximo-distal axis of the cells and showed aberrant clusters. Many of these clusters looked similar to those observed in *w*\* retinas when exposed to constant light (compare Fig 2B' with Fig 1C') but were not present in *w*\* retinas under normal light conditions (Fig 2A'). In addition to large Arl8-positive clusters, crb$^{P13A9}$ mutant retinas frequently displayed even larger patches of Arl8. These appeared to consist of assemblies of vesicles, rather than enlarged individual vesicles (Fig 2C' (arrows) and 2E). It is noteworthy that these patches are present in the *crb*$^{P13A9}$ retina under non-degenerative conditions, and in the absence of any obvious loss of rhabdomeres indicative of the onset of degeneration [32]. The patches were not restricted to photoreceptors (arrow, Fig 2G), but could also be detected in surrounding support cells (arrowheads, Fig 2G).

To characterize the observed differences in the Arl8-positive compartment, we concentrated on the size and intensity of Arl8 spots, since these two features differed the most between mutants and controls, whereas other features, such as aspect ratios and spot numbers, were found not to correlate with any particular genotype. To quantitatively assess the differences, we devised a protocol for detecting and quantifying the Arl8-positive regions in terms of spot size and intensity, using a spot detection script in Fiji (explained in detail in Methods, and see Fig 2C and 2C"). In brief, retinal regions from plane longitudinal sections (judged by Crb staining, green, in Fig 2C) were demarcated manually in the green channel as Regions of Interest (ROIs; Fig 2C and 2C'). Arl8-positive spots were then detected within the ROI for each image ("Arl8 Regions"; beige outlines in Fig 2C"). Sizes and intensities of the Arl8 regions thus defined were then calculated and averaged for each image stack. *crb*$^{P13A9}$ mutant retinas of flies kept under normal 12h/light-dark conditions for 10 days showed a consistent and significant increase in the size and intensity of Arl8-positive regions compared to those of *w*\* mutant retinas (Fig 2H).

## Increase in size and intensity of Arl8-positive compartments in *crb*$^{P13A9}$ mutant retinas depends on light and dietary vitamin A

To test the effect of light on the Arl8-positive compartments in *crb*$^{P13A9}$ mutant retinas, we next modified light conditions by keeping *crb*$^{P13A9}$ flies either in normal light conditions (12h

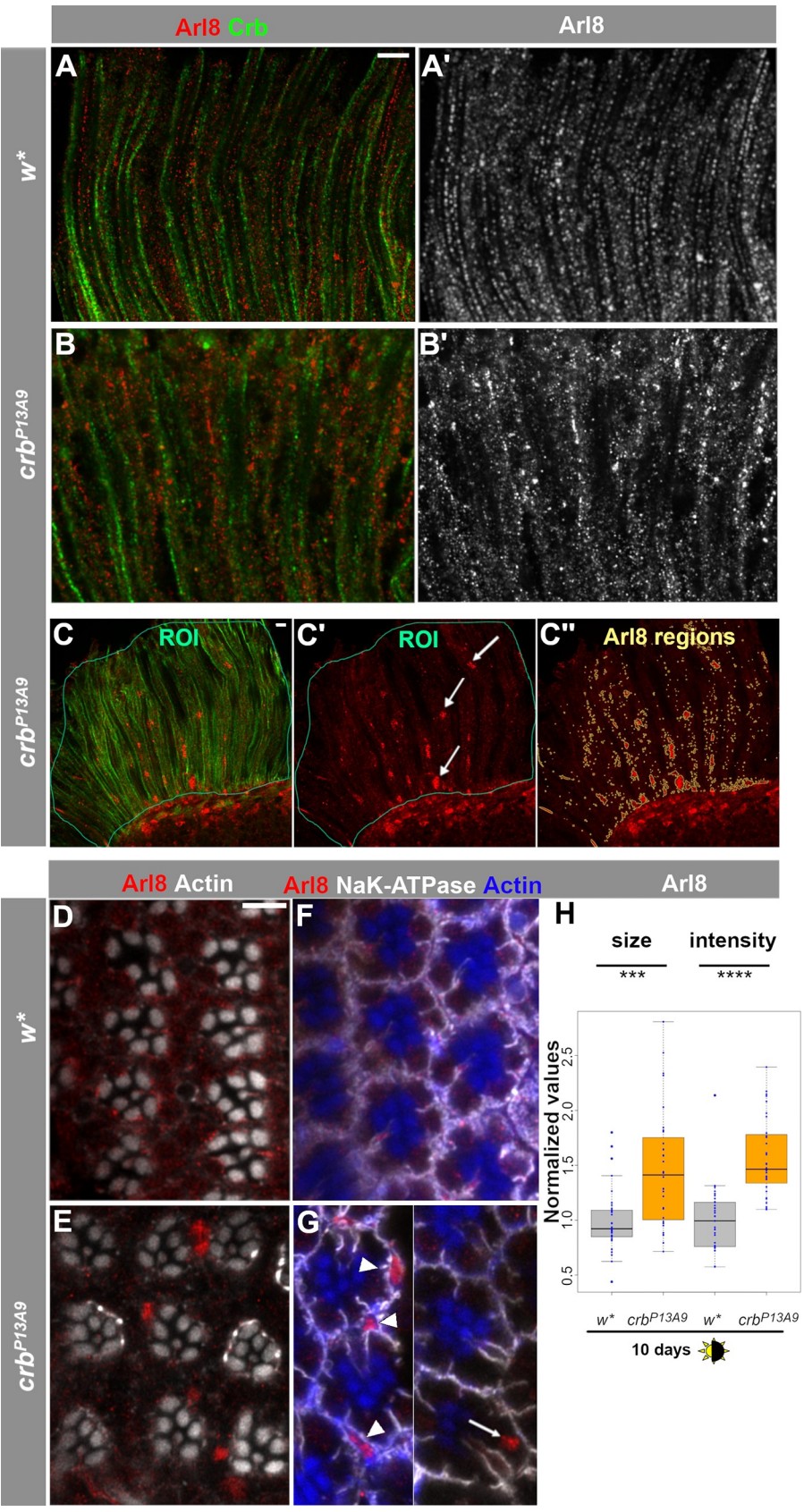

**Fig 2. Arl8-positive lysosomal compartments cluster abnormally in *crb^P13A9* mutant photoreceptor cells under normal light conditions.** (A-B') Optical longitudinal sections of *w\** retinas after 5 days of 12h light/12h dark conditions, stained with anti-Arl8 (red) and anti-Crb (green). (C-C") Depiction of the procedure for quantification of size and intensity of Arl8 regions (see Methods section for details). Areas of longitudinally oriented ommatidia were selected as ROIs (thin teal lines), judged by anti-Crb (green). Arl8-positive regions (red; arrows, C') within the ROIs were determined by an automatically thresholded Find Maxima plugin to detect signal above background, yielding the "Arl8 regions" outlined in beige (C"). (D, E) Optical cross-sections of *w\** (D) and *crb^P13A9* (E) mutant retinas in the same conditions as above show positioning of Arl8 clusters (red) in *crb^P13A9* with respect to rhabdomeres, which are labelled for actin (white). (F, G) Optical cross-sections of *w\** (F) and *crb^P13A9* (G) mutant retinas in the same conditions as above, labelled for basolateral domains (Na⁺-K⁺-ATPase; white) and rhabdomeres (actin; blue), where Arl8-positive clusters (red) can be detected in photoreceptors (arrow) and support cells (arrowheads). (G) is a montage from two different images of *crb^P13A9* rhabdomeres. Scale bar: 5 μm. (H) Comparison of the size and intensity (brightness) of Arl8 regions in *w\** (grey box plots) and *crb^P13A9* (orange box plots) retinas, with averaged data from each image stack shown by a dot (for detailed procedure, see Methods). Under 10 days of normal light/dark conditions, Arl8-positive clusters are detected as larger and brighter regions in *crb^P13A9* retinas (visible in C"), compared to those in *w\** retinas. Size and brightness values were normalized to *w\** controls for both groups, and *w\** and *crb^P13A9* retinas were always processed in parallel in each experiment. Normalized data from up to three experiments were pooled for each box plot (total n for *w\** = 26 image stacks, n for *crb^P13A9* = 24 image stacks). P values for each pairwise comparison were calculated in RStudio using Student's t-test, and are encoded by stars as follows for this and subsequent plots: * P<0.01; **P<0.001; ***P<0.0001; ****P<0.00001; *****P<0.000001.

light/12h dark) or in complete darkness (Fig 3A and 3B'). The differences in size and intensity of Arl8-positive regions observed between *crb* and *w\** after 5 days of normal light-dark conditions (Fig 3E) disappeared when flies were kept in complete darkness for 5 days (Fig 3F; similar results were found after 10 days of complete darkness; not shown).

Retinal degeneration in *crb* mutants is a consequence of light exposure and is rescued in the absence of vitamin A [31, 32], a precursor to the chromophore retinal, which is in turn a component of the light-activatable photopigment rhodopsin. In the absence of vitamin A, Rh1 levels are strongly reduced [46]. Therefore, we asked whether the endolysosomal abnormality reflected by an Arl8 clustering defect in *crb^P13A9* mutants could also be influenced by reducing the amount of Rh1 in the retina. The increase in size and intensity of Arl8-positive clusters in *crb* vs. control retinas observed under normal light conditions was abolished or even slightly reversed when flies were raised and kept in vitamin A depleted food (Fig 3C, 3C' and 3E vs. 3G). To summarize, the observed increase in size and intensity of Arl8-positive compartments in *crb^P13A9* vs. *w\** controls depends on light and rhodopsin.

Because previous studies identified a correlation between the aberrant accumulation of rhodopsin 1 (Rh1) in Rab7-positive compartments and retinal degeneration in the fly [14, 15, 16], we examined whether the aberrant Arl8 clusters in *crb^P13A9* retinas contained Rh1. In retinas of *crb^P13A9* mutant flies raised under normal light conditions, we found that Rh1 accumulated in enlarged intracellular punctae, but these only rarely co-localized with Arl8 (arrowhead, Fig 4B). Large Arl8 clusters and patches in *crb^P13A9* never contained rhodopsin, nor was an extensive overlap seen between Arl8 vesicles and rhodopsin in control retinas (*w\**) under normal light conditions (Fig 4A). Similarly, no localization of Rh1 was found in either genotype under constant intense light conditions (S2 Fig).

## Rab7 is abnormally distributed in *crb^P13A9* and moves to the Arl8 compartment in light

We showed that Arl8 often occupies the same compartment as the late endosomal marker Rab7 in control retinas (S1 Fig) and that the Arl8-positive compartment undergoes changes in morphology and distribution in *crb^P13A9* mutant retinas. Therefore, we next asked whether Rab7 distribution is similarly affected in *crb^P13A9* mutant retinas. No difference in co-localization of Arl8 and Rab7 was observed in control (*w\**) and *crb^P13A9* mutant retinas under normal

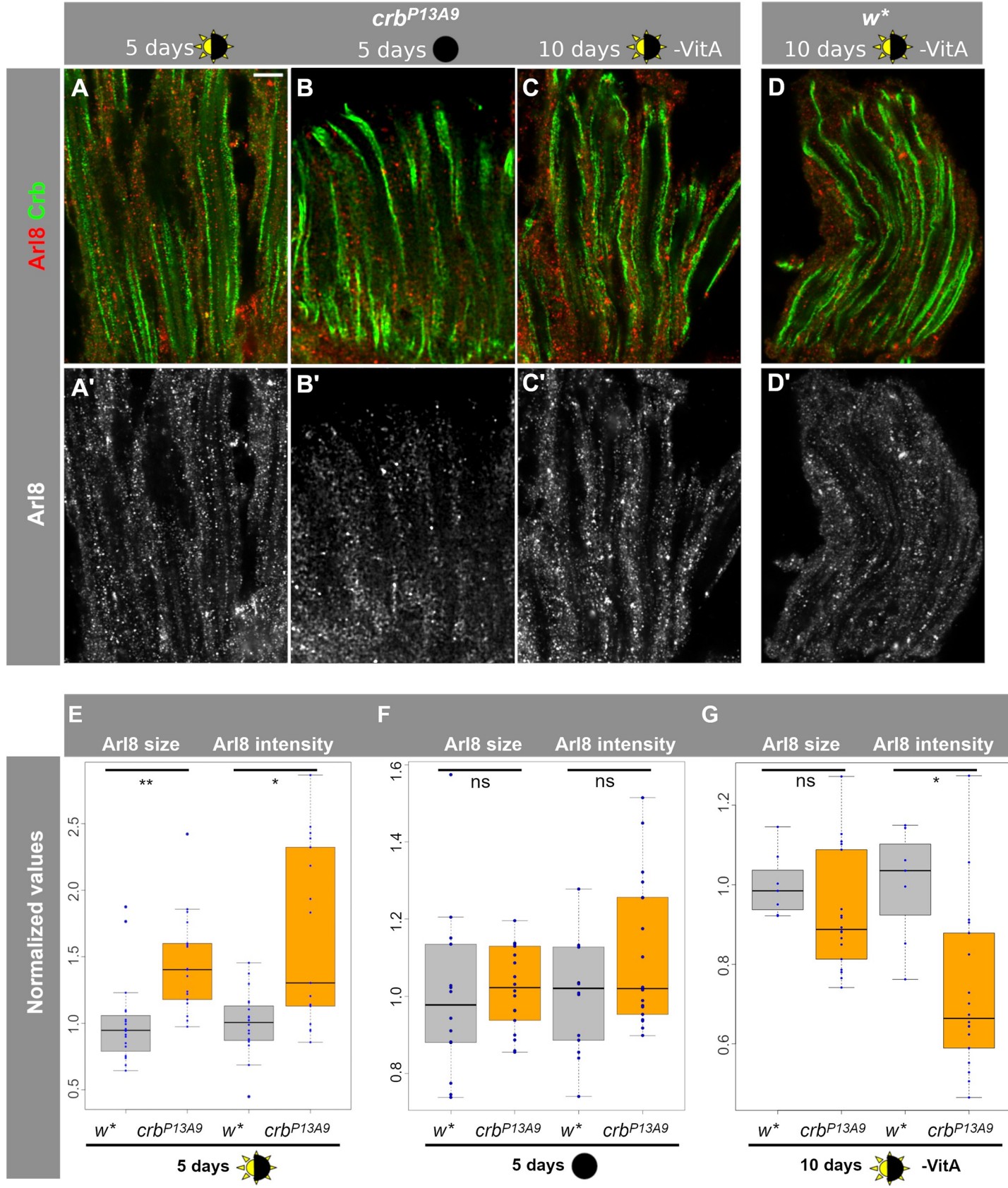

**Fig 3. Arl8 abnormalities in _crb^P13A9_ retinas depend on light and dietary carotenoids.** (A-C') Longitudinal optical sections of _crb^P13A9_ retinas of flies kept for 5 days in 12h light/12h dark (**A, A'**), for 10 days in constant darkness (**B, B'**), or 10 days in 12h light/12h dark without carotenoids (-VitA) (**C, C'**), compared with _w*_ retinas (**D, D'**) kept in the same conditions as (**C**), stained for Arl8 (red; grey in **A'-D'**) and Crb (green). Scale bar: 5 μm. (**E-G**) Quantification of Arl8-positive regions (defined as in Fig 2) shows significant differences in size and intensity between _w*_ (grey box plots) and _crb^P13A9_ (orange box plots) in retinas of flies kept for 5 days in the light conditions indicated below (_w*_ n = 20 image stacks: _crb^P13A9_ n = 17 image stacks). The difference between control and mutant retinas was abolished when flies were raised in constant darkness for 5 days (_w*_ n = 14 image stacks; _crb^P13A9_ n = 18 image stacks) (**F**) or raised for 10 days in light/dark conditions, but without carotenoids in the food (**G**; -VitA) (_w*_ n = 7 image stacks; _crb^P13A9_ n = 18 image stacks). "ns" indicates not significant (P>0.01).

light conditions (Fig 5A and 5C; Fig 5E, brown box plots). Rab7 was observed in some large Arl8-positive clusters of mutant retinas (arrowhead, Fig 5C). When exposed to constant light for two days, the degree of co-localization between Arl8 and Rab7 increased significantly in both genotypes compared to the values determined under normal light conditions (Fig 5B and 5D; Fig 5E, yellow box plots). However, in constant light, total Arl8/Rab7 co-localization was significantly less in _crb^P13A9_ mutant retinas compared to control retinas (Fig 5E, yellow box plots). In addition, large crescent-shaped or hollow vesicles containing both Rab7 and Arl8 were often observed in _crb^P13A9_ mutant retinas upon two days of constant light exposure, but not in control retinas (Fig 5F and 5G). Surprisingly, however, two or six days of constant light exposure did not result in increased size and intensity of Arl8-regions in _crb^P13A9_ mutant retinas relative to _w*_ controls (S3 Fig), as we had observed after five or ten days of normal light exposure (compare to Fig 2H and Fig 3E). To summarize, constant exposure to light promotes

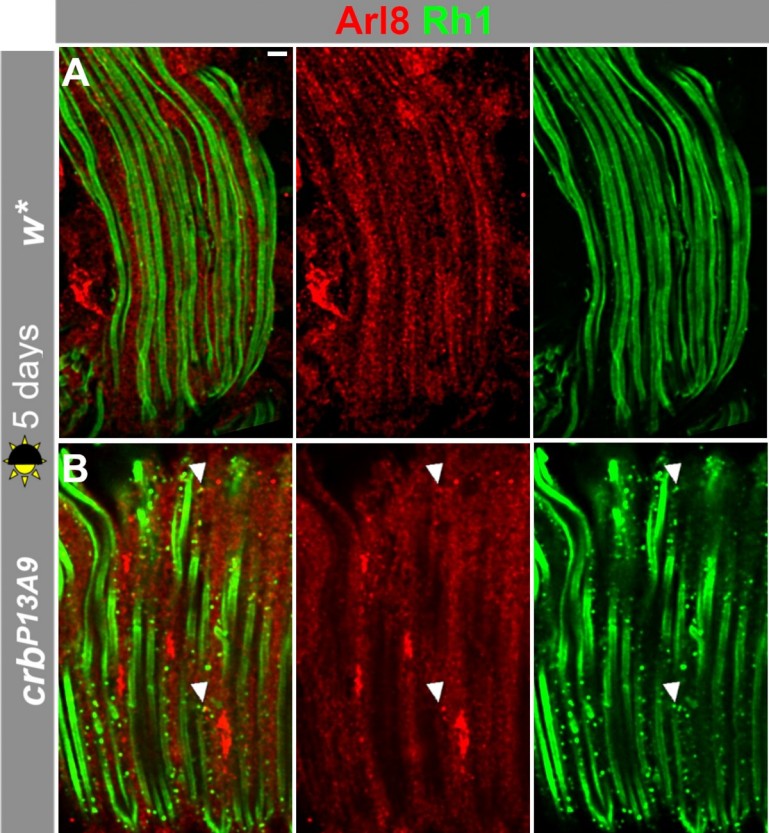

**Fig 4. Arl8- and Rh1-positive compartments are largely non-overlapping in _crb^P13A9_ and _w*_ under normal light conditions.** Longitudinal optical sections of _w*_ (**A**) and _crb^P13A9_ (**B**) retinas of flies kept for 5 days in 12h light/12h dark, stained for Arl8 (red), and Rh1 (green). Note that the mutant retina contains more Rh1-positive vesicles, which are rarely positive for Arl8 (arrowheads). Scale bar: 5 μm.

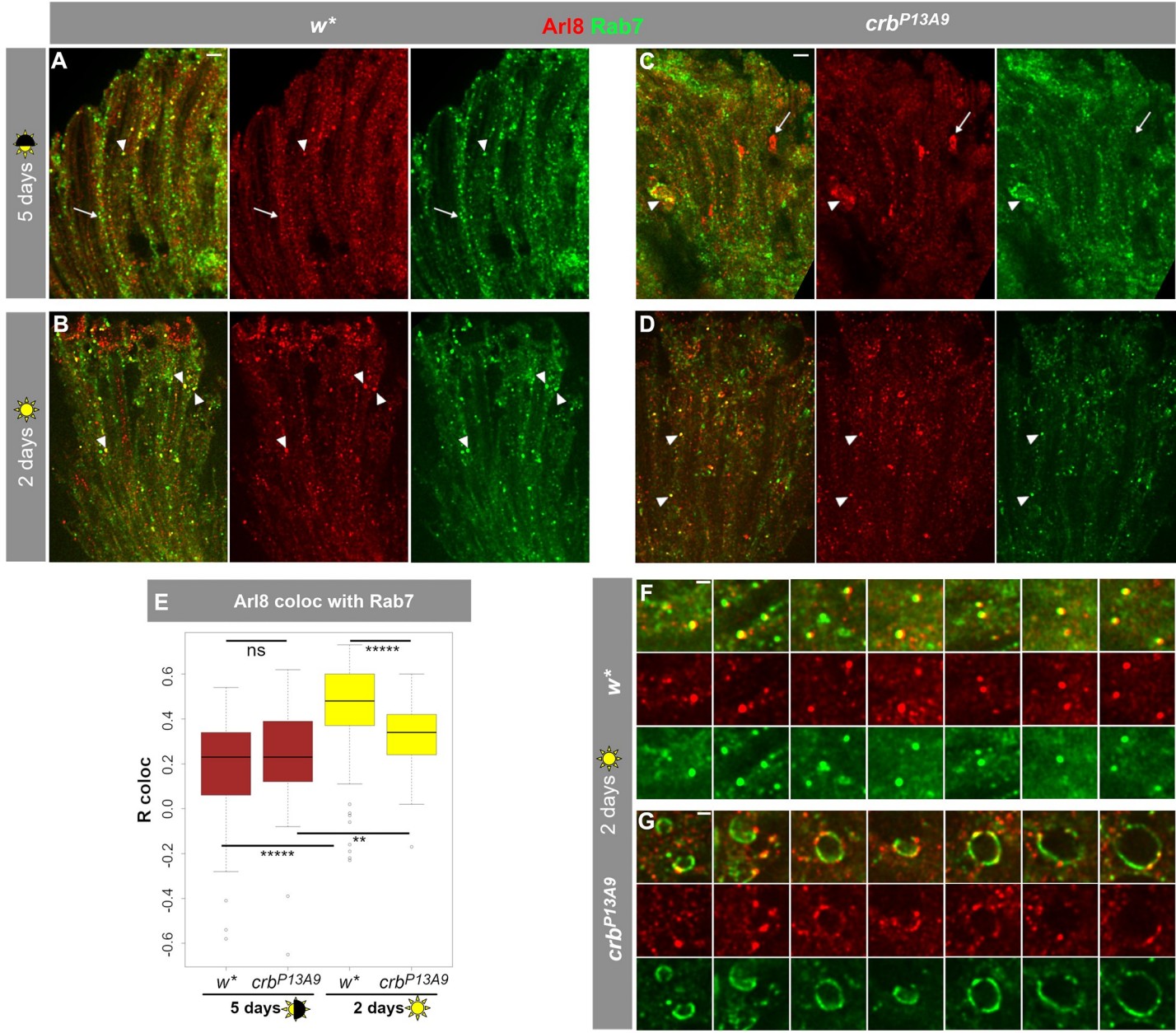

**Fig 5. Exposure to constant light causes increased co-localization of Arl8 with late endolysosomal Rab7 and abnormal Rab7 compartment shapes in *crb*[P13A9]. (A, C)** *w** and *crb*[P13A9] retinas, respectively, under 5 days normal light conditions (12h light, 12h dark) show some overlap of large endogenous Rab7-carrying vesicles (green) with Arl8 (red) (arrowhead). Additionally, many normal-sized compartments in *w** (**A**) and abnormal patches in *crb*[P13A9] (**C**) are non-colocalizing (arrows), resulting in a relatively low, positive Pearson correlation coefficient (R coloc) of ~0.25 (on a scale of -1 to +1) for both genotypes over the whole retina (**E**, brown boxes). **(B, D)** Constant light stress causes an increase in co-localization between Rab7 and Arl8 (arrowheads) in *w** and *crb*[P13A9] retinas, respectively (R coloc = ~0.5 for *w**; ~0.33 for *crb*[P13A9]; **E**). Scale bars for **A-D**: 5 μm. **(E)** Quantification of Rab7 and Arl8 co-localization. 5 days of normal light conditions does not reveal any significant difference in Arl8/Rab7 co-localization (brown box plots). Two days of constant light stress causes a significant increase in the Pearson correlation coefficient in *w**, and a smaller but significant increase in *crb*[P13A9]. Quantification was done on ROIs from image stacks, as described in Fig 2. For 5 days normal light, *w** n = 110 images and *crb*[P13A9] n = 118 images; for 2 days constant light, *w** n = 108 images and *crb*[P13A9] n = 105 images. **(F, G)** Enlarged images showing numerous large round lysosomal compartments in *w** positive for both Arl8 and Rab7. In contrast, large, distended Rab7-positive compartments in *crb*[P13A9] can be seen in contact with small foci of Arl8 (**G**). Scale bars: 1 μm.

co-localization of Arl8 with endogenous Rab7 in control, and to a lesser extent, in *crb*[P13A9] mutant retinas.

### *crb* affects the distribution of the Atg8-positive compartment and its co-localization with Arl8

Arl8 function is closely linked with the autophagosome [49, 50], a compartment that is central to the prevention of retinal degeneration [15]. Since reduction of *crb* induces changes in the Arl8-positive compartment in retinal cells, we monitored possible changes in autophagosomes in the retina of control and *crb*$^{P13A9}$ mutant flies. To do this, we examined Atg8 (Autophagy-related protein 8), which plays a pivotal role during autophagy in the formation of the autophagosomal membrane [51]. We used Atg8-mCherry (a gift from Thomas Neufeld [45]), a marker for the autophagosome and autolysosome. Atg8-mCherry expressed in control photoreceptors using *Rh1*-Gal4 revealed a proximo-distal distribution in photoreceptors (Fig 6A and 6A') under normal light conditions, reminiscent of that of Arl8. Notably, Atg8-mCherry vesicles were adjacent to or overlapping with Arl8-positive vesicles in controls (Fig 6C and 6E). In contrast, Atg8-mCherry-positive compartments in *crb*$^{P13A9}$ mutant photoreceptors clustered (Fig 6B and 6B') and were separated from Arl8 (Fig 6D and 6E). This was reflected in reduced co-localization of individual Atg8-mCherry compartments with Arl8, assessed by measuring Pearson's R coloc and Manders' coefficient for the mCherry channel (tM2), using the Fiji plugin Coloc2 (Fig 6E). These results indicate that reducing *crb* function in photoreceptors lowers the spatial proximity of Arl8 to an Atg8-carrying autolysosomal compartment.

## Discussion

Retinal degeneration is in most cases a gradual process, which raises two questions: i) what are the factors that drive the process of degeneration? And ii) what makes photoreceptor cells carrying mutations in certain genes more prone to degeneration, and how is this predisposition manifested at the cellular level prior to the onset of the degenerative process? In other words, how do extrinsic and genetic, intrinsic factors impair retinal homeostasis? To date, about 100 genes have been linked to retinal degeneration in human [52]. Strikingly, the majority of these have orthologs in *Drosophila* [29]. Many of these genes encode proteins involved in trafficking, which is not surprising, given the high membrane turnover observed in photoreceptors required to maintain homeostasis [53–56]. Although the correlation between a given mutation and the human disease is often well established, we know very little about whether mutations in these genes affect cellular physiology/homeostasis before the onset of degeneration, and if so, how. We hypothesize that such alterations could render photoreceptors more susceptible to external stress factors and hence prone to degeneration.

Using the fly eye as a system to study the genetic control preventing retinal degeneration, we explored the effect of a mutation in *Drosophila crumbs* (*crb*) on the organization of the endolysosomal system in the retina prior to any sign of degeneration. *Drosophila crb* is a well-established model for Leber's congenital amaurosis and RP12-linked retinitis pigmentosa, two severe forms of retinal dystrophy leading to blindness. The recently identified hypomorphic *crb*$^{P13A9}$ allele of *Drosophila* used in this study is particularly suited for these analyses, since unlike other *crb* alleles, it is homozygous viable, shows no developmental retinal phenotype and undergoes retinal degeneration only under constant light stress [32].

The major observations of our study were the following: i) Abnormal clustering of Arl8 was observed in retinas of *crb*$^{P13A9}$ mutant flies under normal light conditions (12 hours dark/12 hours light). Unexpectedly, similar looking clusters also appeared in photoreceptors of *w*$^{*}$ controls, but only under constant light stress. ii) Coincidence of Arl8-positive compartments with late endosomal Rab7 increased under constant light stress in both genotypes (but significantly less so in *crb*). In addition, *crb*$^{P13A9}$ mutant photoreceptors kept in constant light formed large, balloon-like vesicular structures that contained Rab7 on the membrane, but only small foci of

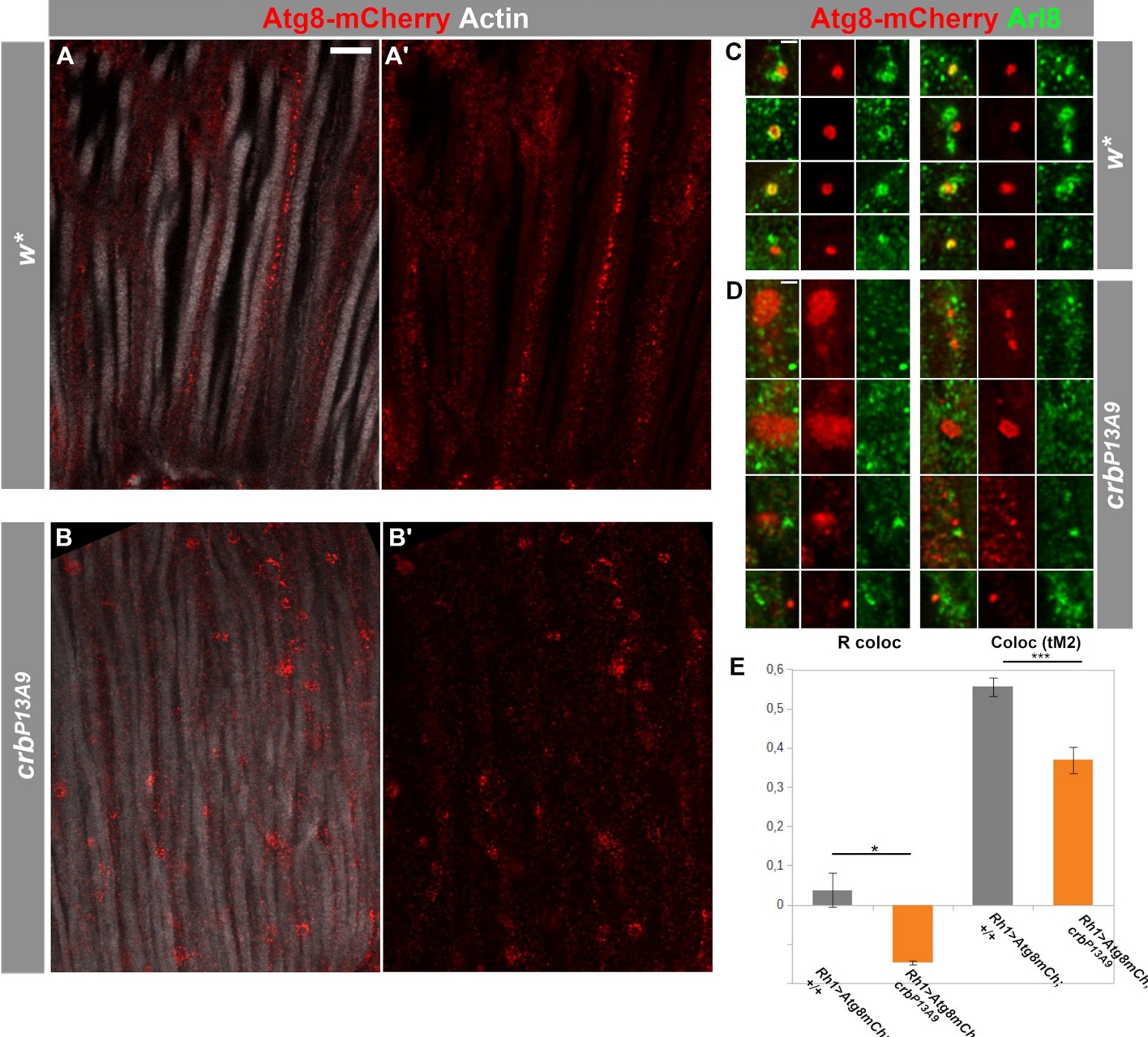

**Fig 6. Autophago-lysosomal marker Atg8-mCherry loses its association with Arl8 in *crb^P13A9* mutant retinas. (A-B')** Longitudinal optical sections of *w\** (**A, A'**) and *crb^P13A9* (**B, B'**) retinas of flies expressing *Rh1*-Gal4-mediated Atg8-mCherry (red). Retinas of flies kept for 6 days at 12h light/12h dark show differences in morphology of autophago-lysosomes in *crb^P13A9* (**B, B'**). Rhabdomeres are labelled for F-actin (phalloidin; white). Scale bar: 5 μm. **(C, D)** Examples of single autophagosomal Atg8-mCherry compartments (red) in *w\** (**C**) and *crb^P13A9* (**D**). Atg8-mCherry is often directly adjacent to or surrounded by Arl8 (green) in the control (**C**), whereas it is non-overlapping with Arl8 in *crb^P13A9* (**D**). Scale bars: 1 μm. **(E)** Quantification showing loss of Atg8-mCherry (driven by Rh1-Gal4) co-localization with Arl8 in *crb^P13A9* in individual compartments, similar to those shown in **C** and **D**) measured by Pearson's R coloc and Manders' thresholded co-localization coefficient tM2 for the Atg8-mCherry channel. Total compartment number (n) for *w\** = 26, n for *crb ^P13A9* = 62.

Arl8. iii) Finally, coincidence of Arl8-positive compartments with autophagosomal Atg8m-Cherry decreased in *crb^P13A9* under normal light conditions. Taken together, data presented here allow us to conclude that retinas with reduced *crb* function exhibit a "pre-degenerative" endolysosomal phenotype, entailing changes in the association of Arl8 with two of its known

interactors, Rab7 and autophagosomes. It should be noted that these defects occur prior to any obvious signs of degeneration, such as rhabdomere loss.

These results raise several questions: How does a reduced function of *crb* influence localization of endolysosomal small GTPases and their associated compartments? To what extent do the morphological alterations of the endolysosomal system affect its function? How do the changes observed make *crb^P13A9* mutant photoreceptors prone to light-induced degeneration?

Degeneration in the vertebrate retina has been subdivided into three phases, with the first phase characterized by increased cell stress due to protein mis-localization (reviewed in [1]). Our data suggest that *Drosophila* retinas mutant for *crb* reveal a pre-degenerative (or degeneration-prone) phenotype when kept under normal light conditions. At least two mechanisms, which have previously been linked to the *crb* mutant phenotype, could promote increased cell stress and hence could be responsible for the induction of such a pre-degenerative state. First, complete loss of *crb* function causes accumulation of intracellular rhodopsin-1 (Rh1)-containing vesicles due to impaired Rh1-transport to the rhabdomere [36]. Defective anterograde trafficking of rhodopsin is a well-established feature associated with retinal degeneration also in the vertebrate retina [56, 57]. Second, loss of *crb* induces an upregulation of NADPH-oxidase activity, resulting in higher levels of reactive oxygen species (ROS) [30]. Elevated ROS levels play a critical role in the pathogenesis of many neurodegenerative diseases, including retinal degeneration [58–61]. There is increasing evidence suggesting a crucial role of ROS in activating autophagy, one of the main degradative pathways critical not only for neuronal health, but also for the survival of other cells, e. g. muscles [59, 62]. During autophagy, engulfed organelles, nutrients, and protein aggregates are delivered to the lysosome to be broken down and degraded or eventually recycled [63–66]. Autophagy is necessary to maintain retinal integrity, and upon its impairment, retinal degeneration occurs in the fly eye, with rhodopsin accumulating in Rab7-positive compartments [15]. However, we did not observe marked co-localization of Rh1-positive vesicles with Arl8 under either normal or light stress conditions in *crb^P13A9* mutant retinas, suggesting that the defects observed, i. e. the modification of the intracellular distribution of the Arl8-, Rab7-, and Atg8-positive compartments, represent another aspect of the mutant phenotype.

Both Arl8 and Rab7 are involved in autophagosome-lysosome fusion, a process that is influenced, among other factors, by the positioning of these compartments [19, 49, 50, 67] (reviewed in [23, 68]). In HeLa cells, for example, lysosomes concentrated in either a perinuclear pool or in the periphery differ in their pH and hence in their functionality [25]. Similarly, the change in positioning of the Arl8- and Rab7-positive compartments under stress (constant light in control flies and 12 h light/ 12 h dark in *crb* mutant flies) and the transformation of the small Rab7-positive compartments into large, balloon-like vesicles with only small foci of Arl8 could indicate a change in their functionality. Likewise, the formation of large Atg8-mCherry clusters and their reduced co-localization with Arl8 in *crb^P13A9* mutant retinas suggest a possible change in function of the autophagosome. Since autophagy is required for degradation of light-activated Rh1 in order to prevent retinal degeneration [15, 69], we tentatively propose that the pre-degenerative state of *crb* mutant retinas is defined by reduced autophagosomal activity, which makes cells prone to degeneration upon increased light stress. Future work will reveal whether the displacement of the Arl8-positive compartments and morphological change in the Rab7-positive vesicles are associated with defects in the degradative machinery, which in turn promotes degeneration [70].

Intracellular positioning and fusion of lysosome depends on both microtubules and the actomyosin network [71]. This raises the question whether Crb might indirectly be involved in regulating lysosomal positioning and function by controlling cytoskeletal organization. In wild-type photoreceptors, Crb protein is restricted to the stalk membrane, a portion of the

apical plasma membrane between the rhabdomere and the adherens junctions [31, 38]. The stalk membrane borders a specialized cortical cytoplasm abutting the base of the rhabdomere, called ectoplasm, which is defined by an actin-rich terminal web also loaded with MyosinV. The ectoplasm is involved in shaping the subrhabdomeral cisternae (SRC), a distinct ER compartment next to the rhabdomere base [72]. Interestingly, proximo-distal alignment of Arl8 in control retinas is just next to the base of the rhabdomere, suggesting a localization at or close to the SRC.

In photoreceptors completely lacking *crb* function or mutant for *crb*[P13A9] the stalk membrane is reduced by about 40% and 20%, respectively [32, 38]. In addition, the amount of $\beta_H$-spectrin at the stalk membrane is strongly reduced in loss of function alleles [38]. The spectrin cytoskeleton recruits microtubule-binding proteins to the apical pole, such as Spectraplakin (called Shortstop in flies) [73, 74]. In this way, the ectoplasm facilitates the movement of cargo, which is required for building the rhabdomere during development and its maintenance in the adult. These cargo include rhodopsin and other molecules destined for the rhabdomeric membrane, but also, notably, pigment granules. Pigment granules are lysosome-related organelles, whose biogenesis and transport pathways share that of lysosomes [40, 75, 76]. In strong light, they align along the rhabdomere base, where they act as a shield from light stress [72]. Since our experiments were performed in a *w* background, the effect of *crb* reduction on pigment granule migration could not be analyzed.

Taken together, we hypothesize that reduced *crb* function results in impaired organization and/or function of the ectoplasm and the SRC. This influences the spatial organization of the endolysosomal system, which, in turn, affects Rab7-Arl8 coordination. These alterations may predispose the hypomorphic *crb* retina toward a degeneration-prone state, which leads to degeneration when subjected to light stress.

## Supporting information

**S1 Fig. The Arl8-positive compartment partially overlaps with late endosomal and autophagosomal markers.** Longitudinal optical sections of *w*[*] retinas after 5 days of 12h light/12h dark conditions, stained with anti-Arl8 (red in all panels) and other lysosomal markers (green in all panels).

**(A, B)** Arl8 occupies the same compartment as endogenous Rab7 (**A**) and *Rh1*-Gal4-mediated Rab7-GFP (**B**) in a sub-population of vesicles. R coloc for Rab7 and Arl8 (shown in Fig 5E) is ~0.17.

**(C)** Arl8-positive compartments are often nearby, but non-identical to those labelled by Atg8-mCherry (green), an autophagosomal marker expressed in all photoreceptors via *Otd*-Gal4. R coloc for Atg8-mCherry and Arl8 (shown in Fig 6E) is ~0.05.

**(D)** Arl8-positive compartments are often nearby, but non-identical with those labelled by the putative lysosomal transporter spinster (spin)-GFP (green), expressed by *Rh1*-Gal4 in photoreceptors R1-6.

**(E)** Arl8-positive compartments are similarly adjacent, but largely non-identical to those labelled by the *bona fide* lysosomal marker, LAMP-GFP, expressed by *Rh1*-Gal4 in photoreceptors R1-6.

**(F)** Arl8 overlaps with a subpopulation of vesicles carrying an intrinsic lysosomal marker, a GFP gene-trap insertion in the $PIP_2$ phosphatase gene (Lyso PIP2-ase), CG6707.

**(G)** Pearson's colocalization coefficient (R coloc) between Arl8 and the markers shown, in retinas. For each stack, R coloc was calculated on regions (ROIs) of every 4th optical slice where clear vesicular staining was present, without obvious background from non-ommatidial tissue. The number of image stacks analysed for each comparison was as follows: Rab7-GFP n = 14

(270 regions); Spin-RFP n = 16 (614 regions); LAMP-GFP n = 13 (408 regions); LysoPIP2-ase n = 12 (233 regions).
Scale bars: 5 μm.
(TIF)

**S2 Fig. Arl8-positive compartments overlap little with Rh1 even under constant light stress.** Longitudinal optical sections of $w^*$ and $crb^{P13A9}$ retinas of flies kept for 6 days in constant light, stained for Arl8 (red) and Rh1 (green). Very few small Arl8-positive compartments are also Rh1-positive (**A-A"**; arrowhead), as is the case in normal light conditions (see Fig 4). In contrast, large Arl8-positive patches in $crb^{P13A9}$ are always negative for Rh1 (**B**).
(TIF)

**S3 Fig. Constant light stress reduces differences in Arl8 compartment size and intensity between $w^*$ and $crb^{P13A9}$.** Arl8 compartments, quantified as in Figs 2 and 3, show no significant differences in size and fluorescence intensity between $w^*$ (grey box plots) and $crb^{P13A9}$ (orange box plots) after two (left graph) or six days (right graph) of constant intense light stress. "ns" indicates not significant (P>0.01). For 2 days constant light, $w^*$ n = 12 image stacks and $crb^{P13A9}$ n = 16 image stacks; for 6 days constant light, $w^*$ n = 20 image stacks and $crb^{P13A9}$ n = 16 image stacks.
(TIF)

**S1 Script. Image analysis script for Fiji that was used to quantify spot regions from spinning disk confocal image stacks.**
(PY)

## Acknowledgments

The authors would like to thank Sarita Hebbar, Ruchi Jhonsa, and Kassiani Skouloudaki for helpful discussions, Nadine Muschalik for the GeneTrap line CG6707, Robert Haase and Benoit Lombardot from the MPI Scientific Computing Facility for the spot analysis script and help with image analysis, Britta Schroth-Diez and the MPI imaging facility for help with spinning disk and confocal imaging, and Jona Drushku for initial screening of compartment markers in retina. This work was supported by the Max-Planck Society. The authors declare that all data and material reported in this manuscript can be made available upon request, and that they have no competing interests.

## Author Contributions

**Conceptualization:** Rachel S. Kraut, Elisabeth Knust.

**Formal analysis:** Rachel S. Kraut.

**Funding acquisition:** Elisabeth Knust.

**Investigation:** Rachel S. Kraut.

**Methodology:** Rachel S. Kraut.

**Project administration:** Elisabeth Knust.

**Resources:** Elisabeth Knust.

**Supervision:** Elisabeth Knust.

**Visualization:** Rachel S. Kraut.

**Writing – original draft:** Rachel S. Kraut.

**Writing – review & editing:** Elisabeth Knust.

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
