## [Decision Letter · Decision Letter 0]

2 Aug 2019

PONE-D-19-19383

Changes in endolysosomal organization define a pre-degenerative state in the crumbs mutant Drosophila retina

PLOS ONE

Dear Dr. Kraut,

Thank you for submitting your manuscript to PLOS ONE. After careful consideration, we feel that it has merit but does not fully meet PLOS ONE’s publication criteria as it currently stands. Therefore, we invite you to submit a revised version of the manuscript that addresses the points raised during the review process.

Both reviewers requested improvements in your statistical analysis, reviewer 2 indicating that the current analysis is inadequate.  Both reviewers requested better explanation of the crumbs protein, and reviewer 1 commented on the high levels of light used in these experiments.

We would appreciate receiving your revised manuscript by Sep 16 2019 11:59PM. To enhance the reproducibility of your results, we recommend that if applicable you deposit your laboratory protocols in protocols.io, where a protocol can be assigned its own identifier (DOI) such that it can be cited independently in the future. For instructions see: http://journals.plos.org/plosone/s/submission-guidelines#loc-laboratory-protocols

We look forward to receiving your revised manuscript.

Kind regards,

Alfred S Lewin, Ph.D.

Academic Editor

PLOS ONE

2. Our internal editors have looked over your manuscript and determined that it is within the scope of our Autophagy and Proteostasis Call for Papers. This collection of papers is headed by a team of Guest Editors: Sharon Tooze, Fulvio Regiori and Thorsten Hoope. The Collection will encompass a diverse range of research articles from early initiation of autophagy, to understand the role other proteostasis pathways play in maintaining cellular homeostasis and the cross talk between the two.  Additional information can be found on our announcement page: https://collections.plos.org/s/autophagy-proteostasis..

If you would like your manuscript to be considered for this collection, please let us know in your cover letter and we will ensure that your paper is treated as if you were responding to this call. If you would prefer to remove your manuscript from collection consideration, please specify this in the cover letter.

Reviewers' comments:

Reviewer's Responses to Questions

**Comments to the Author**

1. Is the manuscript technically sound, and do the data support the conclusions?

Reviewer #1: Yes

Reviewer #2: Yes

2. Has the statistical analysis been performed appropriately and rigorously? 

Reviewer #1: Yes

Reviewer #2: No

3. Have the authors made all data underlying the findings in their manuscript fully available?

Reviewer #1: Yes

Reviewer #2: Yes

4. Is the manuscript presented in an intelligible fashion and written in standard English?

Reviewer #1: Yes

Reviewer #2: Yes

5. Review Comments to the Author

Reviewer #1: Most mutant Crb alleles show a severe retinal morphological phenotype. The manuscript describes a hypomorphic Crb allele that does not show an apparent retinal morphological cell-shape phenotype (although the stalk membrane is 20% reduced), but does show retinal degeneration upon light exposure. Whereas the mutant allele was already previously described in several papers by the authors, here the authors convincingly showed for the first time alterations in endolysosomal compartments before the exposure to light. The authors hypothesize that these changes in the endolysosomal compartments might potentially play a role in the sensitivity of the Crb allele for light stress.

Major questions

1) Add Pearson’s R, mean±SD, p-value and n=samples in the text and each figure legend. Also add this information to the following sentence in the text “Person’s R coloc for individual compartments, was also reduced in crb1(P13A9) (not shown)”.

Minor questions

1) Clarify in the materials and methods that “12 h light, 12 h dark” means “12 h light (2100-2500 lux), 12 h dark (0 lux)”. It is defined under the results section, but put this in the materials and methods. Also, make clear whether or not these light conditions are the same as in previous studies. Also, describe in material and methods the duration that the flies were kept in constant light stress. It also does not sufficiently become clear in the results sections.

Note to the authors: 2100-2500 lux of light is a high dose of light to species that are active at night. In rodent breeding units for example the light levels are often kept below 100 lux. The spectrum of the light might as well play an important role.

Dirk Rieger et al (J Biol Rhythms. 2007 Oct;22(5):387-99; Title: The Fruit Fly Drosophila melanogaster Favors Dim Light and Times Its Activity Peaks to Early Dawn and Late Dusk) described that flies are most active between 5-10 lux and that flies avoid >1000 lux light conditions. So, the flies described in the manuscript might be under 12 h light stress, 12 h dark? In an earlier manuscript of the authors of the current manuscript described that the light exposure as “The total intensity was 17 μmol/m2 s1 of photosynthetically active radiation (380–710 nm) measured by a quantum sensor.” but in that paper (Kevin Johnson et al, 2002) the lux measurements were not described.

2) The procedure for quantification of size and intensity of Arl8 regions is not fully described. It might be useful to add the protocol either in supplementary data or to public websites such as https://ww.protocols.io

3) Whereas the Crb mutant allele P13A9 is described, the manuscript lacks information on how the Crumbs variant protein looks like. How does the Crumbs variant protein look like (what are the changes at amino acid level) and are there similar levels of Crumbs variant protein as in flies expressing the wild type Crumbs protein? Please describe in 2-3 sentences whether or not the Crumbs variant protein localises similar as wild type Crumbs protein. Are there changes in the localisation or levels of interacting proteins in this particular Crb mutant? Whereas the authors sufficiently try, the discussion gives no direct mechanistic insight into how the reduced Crumbs function might result in changes in the localisation of the endolysosomal compartments.

4) Change “Acknowldgements” into “Acknowledgements”

Reviewer #2: In this manuscript, the authors investigate a pre-degenerative state using crbP13A9 mutant that exhibits no developmental and retinal defects in normal light conditions but the onset of retinal degeneration upon light stress. They found that structures, colocalization and distribution of trafficking molecules in the pre-degenerative state were different from those in normal condition. While this manuscript includes interesting findings, I have some significant concerns about specific experiments. The authors would have to fully address these concerns before I would recommend acceptance for publication.

Major points

1) Figure 1B,D : The authors claim that the Arl8-positive compartments are predominantly positioned nearer to the base of the rhabdomere in the normal condition but this positioning was less in constant light condition. However, it is difficult to evaluate this notion by Figure1B and D. Quantitative analysis is needed.

2) Figure S1 : The number, size, intensity and position of Arl8-positive compartments are quite different in these conditions. Thus, I can’t evaluate these data. Quantitative analysis is needed as well.

3) Figure 3F : The authors claim that size and intensity of Arl8-positive compartments are increased by light (Figure 3E) but not change in dark condition (Figure 3F). However, the number of compartments that were tested in Figure 3F is fewer than that in Figure 3E. To raise statistical significance, the number of compartments should be increased for Figure 3F.

4) In all statistical analysis, the number of specimens and compartments tested should be described in FIGURE LEGENDS.

Minor points

1) For non-Drosophila researchers, it should be described what kind of protein Crumb is and what is known about Crumb in INTRODUCTION part.

6. PLOS authors have the option to publish the peer review history of their article (what does this mean?). If published, this will include your full peer review and any attached files.

Reviewer #1: No

Reviewer #2: No

---

## [Author Response · Author response to Decision Letter 0]

23 Oct 2019

Reviewer #1: Most mutant Crb alleles show a severe retinal morphological phenotype. The manuscript describes a hypomorphic Crb allele that does not show an apparent retinal morphological cell-shape phenotype (although the stalk membrane is 20% reduced), but does show retinal degeneration upon light exposure. Whereas the mutant allele was already previously described in several papers by the authors, here the authors convincingly showed for the first time alterations in endolysosomal compartments before the exposure to light. The authors hypothesize that these changes in the endolysosomal compartments might potentially play a role in the sensitivity of the Crb allele for light stress.

Major questions

1) Add Pearson’s R, mean±SD, p-value and n=samples in the text and each figure legend. Also add this information to the following sentence in the text “Person’s R coloc for individual compartments, was also reduced in crb1(P13A9) (not shown)”.

Fig. 5E shows Pearson’s (R coloc) with error bars. Pearson’s R coloc with error bars has been added to fig. 6E, and the text changed accordingly on p.13. Pearson’s R coloc with error bars for other markers with Arl8 was added to fig. S1. The differences here are not quantified because differences in the mode of expression (i.e. Gal4-driven or not) and quality of staining make comparisons difficult. Endogenous Rab7 colocalization and Atg8-mCherry Pearson R colocalization values are already shown in other figures (5 and 5) and so are not repeated in fig. S1.

Minor questions

1) Clarify in the materials and methods that “12 h light, 12 h dark” means “12 h light (2100-2500 lux), 12 h dark (0 lux)”. It is defined under the results section, but put this in the materials and methods. 

Explanation of light conditions has been added to Methods section.

Also, make clear whether or not these light conditions are the same as in previous studies. 

The light conditions are somewhat brighter than the conditions stated in an earlier publication, although the identical incubator was used. 

Also, describe in material and methods the duration that the flies were kept in constant light stress. It also does not sufficiently become clear in the results sections.

“3 days of constant light…” was added to the results section for Fig. 1.

Note to the authors: 2100-2500 lux of light is a high dose of light to species that are active at night. In rodent breeding units for example the light levels are often kept below 100 lux. The spectrum of the light might as well play an important role.

Dirk Rieger et al (J Biol Rhythms. 2007 Oct;22(5):387-99; Title: The Fruit Fly Drosophila melanogaster Favors Dim Light and Times Its Activity Peaks to Early Dawn and Late Dusk) described that flies are most active between 5-10 lux and that flies avoid >1000 lux light conditions. So, the flies described in the manuscript might be under 12 h light stress, 12 h dark? In an earlier manuscript of the authors of the current manuscript described that the light exposure as “The total intensity was 17 μmol/m2 s1 of photosynthetically active radiation (380–710 nm) measured by a quantum sensor.” but in that paper (Kevin Johnson et al, 2002) the lux measurements were not described.

We thank the reviewer for the information on this, and it is indeed interesting to consider. However, for consistency’s sake throughout our experiments, and to avoid the introduction of another variable, we tried to keep the light intensity at the same level throughout (except for dark periods). Empirically, we observed that this relatively high light level (2100-2500 lux) was required for alignment of Arl8 compartments to be observed reliably in w* animals. 

We also note that outdoor light levels on a clear day are much brighter even than in the incubator (~10,000 lux vs. 2500) and an overcast day is ~1075 lux (https://www.engineeringtoolbox.com/light-level-rooms-d_708.html), so even though flies may prefer low light, they are often exposed to similarly high levels. 

2) The procedure for quantification of size and intensity of Arl8 regions is not fully described. It might be useful to add the protocol either in supplementary data or to public websites such as https://ww.protocols.io

The Fiji script that was used for the spot analysis was added to supplemental materials, and referenced in Methods. 

3) Whereas the Crb mutant allele P13A9 is described, the manuscript lacks information on how the Crumbs variant protein looks like. How does the Crumbs variant protein look like (what are the changes at amino acid level) and are there similar levels of Crumbs variant protein as in flies expressing the wild type Crumbs protein? 

Changes in Crb protein isoforms are described on p.10. The isoform Crb_C is not produced (or truncated) but other isoforms are intact and expressed normally in the eye. 

Please describe in 2-3 sentences whether or not the Crumbs variant protein localises similar as wild type Crumbs protein. Are there changes in the localisation or levels of interacting proteins in this particular Crb mutant? Whereas the authors sufficiently try, the discussion gives no direct mechanistic insight into how the reduced Crumbs function might result in changes in the localisation of the endolysosomal compartments.

Regarding the affected isoform Crb_C, see above comment, and detailed information about the allele in Spannl et al (referenced in the paper). We propose a possible explanation for the endolysosomal defects on p.17: “…we hypothesize that reduced crb function results in impaired organization and/or function of the ectoplasm and the SRC. This influences the spatial organization of the endolysosomal system, which, in turn, affects Rab7-Arl8 coordination.” This is stated after a discussion of the known interaction of Crb with the cytoskeleton, and its location directly abutting the SRC (subrhabdomeric cisternae). 

4) Change “Acknowldgements” into “Acknowledgements”

This has been corrected. 

Reviewer #2: In this manuscript, the authors investigate a pre-degenerative state using crbP13A9 mutant that exhibits no developmental and retinal defects in normal light conditions but the onset of retinal degeneration upon light stress. They found that structures, colocalization and distribution of trafficking molecules in the pre-degenerative state were different from those in normal condition. While this manuscript includes interesting findings, I have some significant concerns about specific experiments. The authors would have to fully address these concerns before I would recommend acceptance for publication.

Major points

1) Figure 1B,D : The authors claim that the Arl8-positive compartments are predominantly positioned nearer to the base of the rhabdomere in the normal condition but this positioning was less in constant light condition. However, it is difficult to evaluate this notion by Figure1B and D. Quantitative analysis is needed.

We agree with the criticism, and indeed tried to quantitate the positioning of the Arl8 compartments with respect to the rhabdomere. However, we found this to be very problematic: we were unable to set a threshold intensity or size for the spots that could correctly exclude other nearby Arl8 spots and identify the aligned spots. In cross-sections, the proximity of neighboring rhabdomeres also makes this difficult. We considered trying to quantitate the position in 3D reconstructions of the entire ommatidia, but were advised by our computer imaging specialists that this would be very difficult or impossible. We slightly changed the wording of the description on p.9 to remove any suggestion of a quantitative change, since we were limited to subjective observation. 

2) Figure S1 : The number, size, intensity and position of Arl8-positive compartments are quite different in these conditions. Thus, I can’t evaluate these data. Quantitative analysis is needed as well. 

A quantitative analysis of Pearson’s R has been added to fig. S1. Due to differences in background levels and mode of expression for the different markers, we did not calculate whether the differences in R coloc were significant, but the results confirmed our overall impression of the extent of colocalization with Arl8. 

3) Figure 3F: The authors claim that size and intensity of Arl8-positive compartments are increased by light (Figure 3E) but not change in dark condition (Figure 3F). However, the number of compartments that were tested in Figure 3F is fewer than that in Figure 3E. To raise statistical significance, the number of compartments should be increased for Figure 3F.

4) In all statistical analysis, the number of specimens and compartments tested should be described in FIGURE LEGENDS.

We have updated the figure with a new analysis for the dark condition, with more stacks. We observed an increase of size/intensity of Arl8 compartments in crb compared to w* in 5 days of 12h light/12h dark (Fig. 3E), and this difference was reduced to insignificance in 5 days dark (Fig. 3F). The original 10 days light/dark graph has been moved to Fig. 2. 

Minor points

1) For non-Drosophila researchers, it should be described what kind of protein Crumb is and what is known about Crumb in INTRODUCTION part.

A short description of the Crb protein has been added to the introduction, p. 4-5.

---

## [Decision Letter · Decision Letter 1]

25 Nov 2019

Changes in endolysosomal organization define a pre-degenerative state in the crumbs mutant Drosophila retina

PONE-D-19-19383R1

Dear Dr. Kraut,

We are pleased to inform you that your manuscript has been judged scientifically suitable for publication and will be formally accepted for publication once it complies with all outstanding technical requirements.

With kind regards,

Alfred S Lewin, Ph.D.

Section Editor

PLOS ONE

Additional Editor Comments (optional):

Reviewers' comments:

Reviewer's Responses to Questions

**Comments to the Author**

1. If the authors have adequately addressed your comments raised in a previous round of review and you feel that this manuscript is now acceptable for publication, you may indicate that here to bypass the “Comments to the Author” section, enter your conflict of interest statement in the “Confidential to Editor” section, and submit your "Accept" recommendation.

Reviewer #2: All comments have been addressed

Reviewer #3: (No Response)

2. Is the manuscript technically sound, and do the data support the conclusions?

Reviewer #2: Yes

Reviewer #3: Yes

3. Has the statistical analysis been performed appropriately and rigorously? 

Reviewer #2: N/A

Reviewer #3: Yes

4. Have the authors made all data underlying the findings in their manuscript fully available?

Reviewer #2: Yes

Reviewer #3: Yes

5. Is the manuscript presented in an intelligible fashion and written in standard English?

Reviewer #2: Yes

Reviewer #3: Yes

6. Review Comments to the Author

Reviewer #2: I understand that it is difficult to quantify the data presented in Figure 1. The other comments are addressed satisfactory.

Reviewer #3: The authors use a viable crb allele that leads to retinal degeneration upon light illumination, to study whether and how defects in crb function might lead to defects in the endolysosomal pathway, which is often aberrant degenerating neurons. They find that alterations in endolysosomal compartments precede exposure to light. They propose that these alterations could play a role in photoreceptor degeneration upon light illumination. Overall, the data presented in the paper are clear and support the authors conclusions. This is a very interesting paper and I think that it should be published as it is.

7. PLOS authors have the option to publish the peer review history of their article (what does this mean?). If published, this will include your full peer review and any attached files.

Reviewer #2: No

Reviewer #3: No

---

## [Editor Report · Acceptance letter]

6 Dec 2019

PONE-D-19-19383R1 

Changes in endolysosomal organization define a pre-degenerative state in the *crumbs* mutant *Drosophila* retina 

Dear Dr. Kraut:

I am pleased to inform you that your manuscript has been deemed suitable for publication in PLOS ONE. Congratulations! Your manuscript is now with our production department. 

With kind regards,

on behalf of

Dr. Alfred S Lewin 

Section Editor

PLOS ONE